# Stability Studies of Clonazepam 2.5 mg/mL Oral Solution and 1 mg/mL Parenteral Solution in Pre-Filled Polypropylene Syringes

**DOI:** 10.3390/pharmaceutics17101302

**Published:** 2025-10-07

**Authors:** Juan Carlos Ruiz Ramirez, Icram Talsi Hamdani, Laura Bermúdez Gazquez, Alice Charlotte Viney, José M. Alonso Herreros

**Affiliations:** 1Deparment of Pharmacology, Faculty of Pharmacy, University of Murcia, 30003 Murcia, Spain; icram95@gmail.com (I.T.H.); laurabermudezgazquez@gmail.com (L.B.G.); josem.alonso@carm.es (J.M.A.H.); 2Pharmacy Service, Los Arcos del Mar Menor General University Hospital, Murcian Health Service, 30739 Murcia, Spain; alice.charlotte@carm.es; 3Biomedical Research Institute of Murcia Pascual Parrilla—IMIB, 30120 Murcia, Spain

**Keywords:** clonazepam, stability study, pre-filled syringes, validation, HPLC

## Abstract

**Background:** Clonazepam is a benzodiazepine drug indicated in all clinical forms of epileptic seizures, various forms of myoclonic seizures, myoclonus and other abnormal movements. At present, it is classified as a hazardous drug requiring special precautions for personnel at reproductive risk, according to a technical document produced by the Spanish National Institute for Safety and Health at Work (INSST), in collaboration with the Spanish Society of Hospital Pharmacy (SEFH). The commercial solutions of clonazepam, for oral and parenteral administration, are supplied by laboratories in glass containers. Repacking in pre-filled polypropylene (PP) syringes, made in the pharmacy service, and in aseptic conditions, may facilitate its administration and reduce the risks to the health or safety of nursing personnel. Nevertheless, there is a lack of stability studies of clonazepam in pre-filled PP syringes. **Objectives**: To evaluate the physicochemical stability of commercial clonazepam 2.5 mg/mL oral solution and 1 mg/mL parenteral solution repackaged in pre-filled PP syringes under various storage conditions. **Methods**: A rapid, linear, precise and sensitive high-performance liquid chromatography (HPLC) method for chemical stability studies of Clonazepam 1 mg/mL (parenteral use) and 2.5 mg/mL (oral use) in solution was implemented after repackaging in pre-filled PP syringes. The studies were conducted by measuring concentrations of oral and parenteral clonazepam in pre-filled syringes, at various time points, over 30 days in several different storage conditions: oral clonazepam protected from light in refrigerator and at controlled room temperature exposed to ambient light; parenteral clonazepam protected from light in a refrigerator and at controlled room temperature protected or unprotected from light. Visual aspects and pH change as well as crystal formation were checked to determine physical stability. **Results**: The degradation of the active ingredient in all groups was less than 10% after 30 days. No evidence of crystal formation, pH and visual aspect changes were observed. **Conclusions**: Clonazepam 1 mg/mL parenteral solution and 2.5 mg/mL oral solution in pre-filled PP syringes are stable for up to 30 days in the tested conditions. The centralized repackaging of clonazepam in pre-filled PP syringes, connected to a closed safety system, in the pharmacy service, reduces drug manipulation by nursing staff decreasing the risk of occupational exposure.

## 1. Introduction

Clonazepam, 5-(2-chlorophenyl)-1,3-dihydro-7-nitro-2H-1,4-benzodiazepin-2-one is a drug that belongs to the group of benzodiazepines (Figure 1). Its mechanism of action involves allosteric interactions between central benzodiazepine receptors and gamma aminobutyric acid (GABA) receptors in the brain, enhancing the effects of GABA. In Spain, it is indicated in most of the clinical forms of epileptic disease and seizures in infants, children and adults. In the last group, it is used in status epilepticus too [1,2].

The aromatic nitro group, the cyclic amide, and the covalently bonded chlorine atom within the clonazepam molecule can influence its chemical stability, both by environmental factors and storage conditions, while also impacting its pharmacological properties and toxicological profile. The nitro group may be susceptible to reduction reactions, potentially affecting the drug’s toxicity and safety. Under extreme conditions, it could contribute to the formation of toxic degradation products. The cyclic amide imparts molecular rigidity, enhancing resistance to chemical degradation and hydrolysis; however, under extreme temperature and pH conditions, this cyclic group may break, facilitating the degradation of the active pharmaceutical ingredient. The covalently bonded chlorine atom is chemically stable, conferring additional chemical resistance, although it can serve as a reactive site in photochemical or oxidative degradation processes; moreover, it contributes to increasing the molecule’s lipophilicity, thereby affecting its metabolism and distribution [1,2].

The INSST and the National Institute for Occupational Safety and Health of the United States (NIOSH) include clonazepam in their List of Hazardous Drugs in Healthcare Settings and classify it as a group 3 non-antineoplastic drug that primarily has adverse reproductive effects. These risks affect men and women of reproductive age, pregnant women or those who are breastfeeding, but do not pose a significant risk to other personnel [3,4]. The FDA classified clonazepam as a category “D” pregnancy risk drug prior to 2015 [5]. According to NIOSH and INSST guidelines, special protection is not required for handling clonazepam except for personnel at reproductive risk (pregnancy, breastfeeding, or those actively trying to conceive). For this risk group, it is recommended to handle the drug with double gloves, gown, face mask and eye protection as a safeguard against the risk of splashes or inhalation.

In Spain, there are two presentations of clonazepam commercialized in solution: oral drops (2.5 mg/mL) and parenteral (1 mg/mL); both are used in our hospital. For the reasons stated above, the hospital’s Pharmacy and Therapeutic Commission, together with the Occupational Risk Prevention Service, proposed the possibility of dispensing the two clonazepam solutions in pre-filled syringes, prepared under aseptic conditions at the pharmacy service, because a pre-filled syringe is a ready-to-use system that decreases the hazards of drug manipulation and also saves nursing time.

After evaluating the request from a pharmaceutical point of view, the work would consist of repackaging these drugs (the oral and the parenteral solution) from their original packaging into a PP syringe. The oral solution is packaged in a topaz glass bottle and the parenteral in a topaz glass ampoule. The conservation conditions of two presentations are temperature equal to less than 30 °C, in its secondary packaging to protect from the light. In these conditions, both have a three-year shelf life if unopened [1,2]. The issues to be clarified would therefore be how long these drugs are physicochemically stable, in PP syringes, at concentrations of 1 mg/mL and 2.5 mg/mL.

PP is considered one of the most inert plastics with the lowest tendency to adsorb drugs; in practice, adsorption to PP is usually low, but it can be significant in drugs of low concentration, of protein or peptide nature, or of high unit value, such as certain radiopharmaceuticals or insulin. In addition, many studies have shown that the physicochemical stability of most drugs is adequate in PP syringes, with no appreciable degradation over periods ranging from several days to weeks, depending on the drug and the storage conditions (refrigeration, darkness). PP offers low permeability to water vapor and is relatively impermeable to oxygen, which helps protect the solution from degradation by oxidation or hydrolysis. However, its gas barrier is not as hermetic as that of glass, so in long-term storage situations the shelf life of very sensitive solutions could be affected. Pharmaceutical-grade PP syringes minimize the presence of particles and the migration of additives [6,7,8]. Therefore, the stability of clonazepam solutions (2.5 mg/mL and 1 mg/mL) could be affected by PP under different conditions. The relationship between the concentration of a drug in solution and its stability is not generally directly proportional, but rather depends on multiple chemical, physical, and environmental factors surrounding the drug. The chemical stability of a drug in solution can be affected by concentration, but the relationship is not always linear or direct. In certain cases, higher concentrations can favor stability by reducing the proportion of water, but in other cases, they can increase degradation due to chemical or physical reactions that depend on the active ingredient concentration and on other factors such as pH, temperature, and the presence of ions or catalysts. Stability also depends on degradation kinetics. This determines how the drug concentration changes over time and how it degrades, regardless of whether its concentration is low or high. For example, in some cases the degradation rate is proportional to the concentration, and in others it is not. For these reasons, the stability must be specifically studied for each formulation, as behavior can vary depending on the drug and its environment [9,10,11,12,13,14].

Nevertheless, the current lack of stability studies of clonazepam in pre-filled PP syringes prevents the pharmacy services from preparing and storing it. Only one study of clonazepam solution 0.5 mg/mL in water for injection, packaged in PP syringes has been published, with the result of 48 h of stability, at 25 °C, not protected from light [15].

Therefore, this study investigates the physicochemical stability of clonazepam in pre-filled PP syringes in several different conditions.

## 2. Materials and Methods

### 2.1. Sample Preparation of Pre-Filled Syringes

Oral clonazepam 2.5 mg/mL solution syringes: Amber polypropylene 1 mL light protected oral syringes (Becton DickinsonTM, Madrid, Spain) with a tip cap were pre-filled with 0.4 mL of clonazepam 2.5 mg/mL oral solution drops (Rivotril^®^, Roche Farma, S.A., Madrid, Spain). Each syringe contained 1 mL of clonazepam (Figure 2). Two groups of syringes were stored at either controlled room temperature (25 °C) exposed to ambient light, or under refrigerated conditions (2–8 °C) protected from light.

Parenteral clonazepam 1 mg/mL solution syringes: Luer lock polypropylene syringes (Nipro Europe Group Companies, Madrid, Spain) for parental use were pre-filled with 1 mL of clonazepam 1 mg/mL parenteral solution (Rivotril^®^ powder 1 mg + 1 mL solvent, Roche Farma, S.A., Madrid, Spain), connected to a closed safety system (TexiumTM, Becton Dickinson España, S.A., Madrid, Spain) as shown in Figure 3. Two groups of syringes were stored at either controlled room temperature (25 °C) protected and unprotected from light, or under refrigerated conditions (2–8 °C) protected from light.

### 2.2. Chemical Stability

The chemical stability of oral and parenteral clonazepam in pre-filled syringes was studied over 30 days of storage (day 0; days 1 to 4; days: 7, 9, 11, 14, 17, 21, 24, 28 and 30).

The chemical stability was studied on the selected days by withdrawing an aliquot of each syringe that was then diluted with the mobile phase to a concentration of 25 µg/mL. Three different batches of each preparation were analyzed in triplicate by HPLC within 10 min of dilution.

If the drug concentration remained between 90% and 110% of the initial concentration during the 30 days of storage, the preparation was considered stable [9,14,16].

### 2.3. Chromatographic Method

A Waters Breeze HPLC system (Waters Cromatography, S.A., Barcelona, Spain) and a XBridge 5 µm C18 (130 Å pore size, 4.6 × 150 mm) reversed-phase column (Waters Cromatography, S.A., Barcelona, Spain) were used. The chromatographic conditions were isocratic mobile phase composed of ultrapure water/acetonitrile/methanol (40/30/30 *v*/*v*) at a flow rate of 1 mL/min, ultraviolet detector at 254 nm, 30 °C column temperature, injection volume of 20 μL and run time of 5 min [17,18]. The HPLC reagents acetonitrile (HPLC-grade) and methanol (HPLC-grade) were purchased from Panreac Química S.L.U. (Barcelona, Spain). The reference drug clonazepam was obtained from Roche Farma, S.A. (Madrid, Spain).

Validation of the method: The HPLC method was validated in terms of linearity, precision and accuracy according to the ICH guidelines [19]. Linearity: Linearity between the peak area and the clonazepam concentration was evaluated by performing six measurements in a concentration range of 0–45 μg/mL (0, 6, 15, 24, 30 and 45 μg/mL). A calibration curve and the corresponding linear regression analysis were performed, obtaining the results of the coefficient of determination (R^2^), the slope (a) and the Y intercept (b) [16]. Precision: Precision was verified by repeatability in intra and inter-day studies. The intra-day study consisted of analyzing five times, on the same day, with the samples at 80, 100 and 120% of the target concentration (25 µg/mL). In the inter-day study the samples were analyzed two times during four different days, at 80, 100 and 120% of the target concentration. The mean, the standard deviation and the coefficient of variation were calculated, with less than 1% variation being accepted for intra-day repeatability, and less than 2% for inter-day repeatability [16,20]. Accuracy: Accuracy was determined by recovery studies in triplicate at 20, 25 and 30 μg/mL concentrations of clonazepam. Recovery was expressed as a percentage, and the mean value was compared with the theoretical value (100%), using Student’s *t* test [16,20,21]. Limit of detection and limit of quantification: To determine the limit of detection (LOD) and the limit of quantification (LOQ) of clonazepam, the independent term “b” and the slope “a” were used in the equations LOD = 3 b/a and LOQ = 10 b/a [16,22].

This chromatographic method had been previously validated in a published study, which assessed its stability by analyzing clonazepam samples subjected to heat, light, acidic, basic, and oxidative degradation. The study demonstrated that the presence of degradation products did not interfere with the detection of the intact drug [17].

### 2.4. Physical Stability

Physical stability was studied by checking visual aspects, determining pH and observing crystal formation. Further, 1 mL samples of each preparation were obtained on the selected evaluation days and were checked for visual aspects such as particle formation, crystals, turbidity, precipitation, color changes and leakage or blockage of syringe plunger during storage. The pH was measured with a calibrated SevenMultiTM pH meter (Mettler Toledo, Cornellà de Llobregat, Spain). A visual inspection booth with both a black/white background [12] and bright-field, and a SediMAX2TM phase contrast microscope (77 Elektronika, Budapest, Hungary), were used to determine the presence of particles and crystals.

## 3. Results

### 3.1. Validation of the Analytical Method

The method demonstrated excellent linearity, with a R^2^ greater than 0.9996. The regression equation was calculated as y = 127,436x + 8625 (Appendix A). The results were highly satisfactory regarding the intra-day and inter-day repeatability of the three clonazepam quality control solutions (Appendix A). Accuracy ranged from 98.34% to 101.62% (Appendix A), while precision, expressed as relative standard deviation (RSD%), fell within the range of 0.094% to 0.682% for intra-day precision and 0.232% to 0.713% for inter-day precision. These RSD% values comfortably meet the ICH (International Conference on Harmonisation) standards, which stipulate a maximum RSD of 1% and 2%, respectively. Likewise, the accuracy was between 98% and 102%. The LOD and the LOQ were calculated as 0.20 µg/mL and 0.68 µ/mL, respectively (Table 1 and Table 2). 

### 3.2. Stability Study

The stability study was carried out by measuring the concentration of clonazepam in pre-filled syringes on each day of the analysis, as described previously. The mean concentrations were calculated and expressed as a recovery percentage with respect to the measurement on the first day (D0 = 100%). The chromatograms of D0 and D30 (day 30 of the study) for each preparation under the specified storage conditions are presented in Figure 4. The clonazepam 1 mg/mL parenteral solution contains benzyl alcohol as a preservative agent. This compound exhibits absorption at 254 nm, which accounts for the appearance of a corresponding peak in the chromatograms.

The results show that the concentrations remained stable for 30 days in all of the storage conditions used for the oral clonazepam 2.5 mg/mL solution syringes (Table 3 and Figure 5) and the parenteral clonazepam 1 mg/mL solution syringes (Table 4 and Figure 6). For more detailed information, the data obtained for each of the days on which concentration monitoring was performed can be consulted in Appendix A (Appendix A).

No significant variation was observed with regard to visual aspects (color changes, turbidity) and pH throughout the study (Table 3 and Table 4).

## 4. Discussion

Since the publication of the document on hazardous drugs and preventive measures for their preparation and administration, Hospital Pharmacy Services have been involved in implementing measures to adapt practices to the recommendations given by the INSST and European Agency for Safety and Health at Work in its guidance document for the safe management of hazardous medicinal products at work [23]. Among the adaptation options is the possibility of direct delivery of the drug from the pharmacy service in a standardized dose and container, ready for administration. If stability studies of the drug in standardized containers are available, it is possible to prepare and store the drug in at pharmacy service, depending on consumption, to avoid the need for shift or daily preparation.

The HPLC method used for the determination of clonazepam was linear, precise, accurate, repeatable, and reproducible, and it was validated in accordance with ICH guidelines. This method was demonstrated to be robust and capable of reliably resolving clonazepam in the presence of its degradation products [17], consistent with the findings of other published studies [20,24].

No studies have evaluated the degradation kinetics of clonazepam in an oral solution at a concentration of 2.5 mg/mL or in a parenteral solution at 1 mg/mL, both packaged in polypropylene syringes. One study assessed the stability of an acidic clonazepam solution at 2 mg/mL in glass ampoules, concluding that clonazepam degradation follows first-order kinetics [25]; similarly, another study investigated the stability of clonazepam tablets in combination with escitalopram [24]. As already mentioned in the introduction, both clonazepam concentrations analyzed in this study have a manufacturer-reported shelf life of three years when stored in primary glass containers under the recommended storage conditions, suggesting that both are highly stable. The aim of this study was not to determine whether repackaging commercial clonazepam solutions in polypropylene syringes could affect or alter the degradation kinetics of the active ingredient, although conducting such an investigation would be advisable to further strengthen the conclusions.

There are currently only two studies published that evaluate the stability of oral clonazepam solutions. One of them assessed the stability of clonazepam 0.2 mg/mL oral solution stored under refrigeration (2–8 °C) and at room temperature, using clonazepam in powder form to prepare the solution and pack on amber-colored glass bottles, concluding that the solution was stable for 90 days [18]. The other analyzed the stability of clonazepam 0.1 mg/mL oral solution prepared from commercial tablets, placed in polyethylene terephthalate bottles, both stored under refrigeration (2–8 °C) and at room temperature protected from light, observing that the solution remained stable for 60 days under both storage conditions [17]. To date, only one study has investigated the stability of clonazepam in polypropylene syringes, although at a lower concentration (0.5 mg/mL). Unlike the previous cases, this study has evaluated the stability of commercialized clonazepam drugs in the presentations of 2.5 mg/mL oral drops and 1 mg/mL injectable solution, repackaged in pre-filled PP syringes. Until this study was carried out, the stability of these commercials solutions were known in glass containers but not in polypropylene containers. Both concentrations are significantly higher than the concentrations mentioned in the studies beforehand, a condition that does not seem to affect the stability observed in the current work.

No significant changes in the pH, or in the appearance or coloration of the analyzed oral and parenteral solutions were observed in this study. No particle formation, crystals, turbidity, precipitation or leakage or blockage of the syringe plunger during storage were observed, regardless of the storage conditions. These results were interpreted as indicative of the stability of the two solutions.

It is considered that performing X-ray diffraction analysis or differential scanning calorimetry to determine whether the crystalline state of the drug was maintained or altered during storage would have provided greater consistency to the results, and these techniques were unfortunately not available at the time of the study.

The assessment of the microbiological stability of commercial clonazepam solutions repackaged in polypropylene syringes was not within the objectives of this study. Nevertheless, according to the manufacturer’s product information, the oral solution at a concentration of 2.5 mg/mL has a shelf-life of 120 days after opening. Regarding the 1 mg/mL solution intended for parenteral administration, it should be noted that it contains benzyl alcohol as a preservative. Therefore, since the repackaging process was carried out under aseptic conditions in a laminar airflow cabinet, the stability of both commercial solutions was not considered to be compromised.

## 5. Conclusions

In the present study, clonazepam 2.5 mg/mL oral solution in light-protected pre-filled PP syringes, both at room temperature and under refrigeration (2–8 °C), and clonazepam 1 mg/mL parenteral solution in pre-filled PP syringes at room temperature with and without light protection, and under refrigeration (2–8 °C) with light protection, are observed to be physically and chemically stable for at least 30 days. This allowed for the preparation of ready-to-use stock of this hazardous drug, minimizing drug manipulation required from nursing staff and therefore reducing the risk of occupational exposure.

## Figures and Tables

**Figure 1 pharmaceutics-17-01302-f001:**
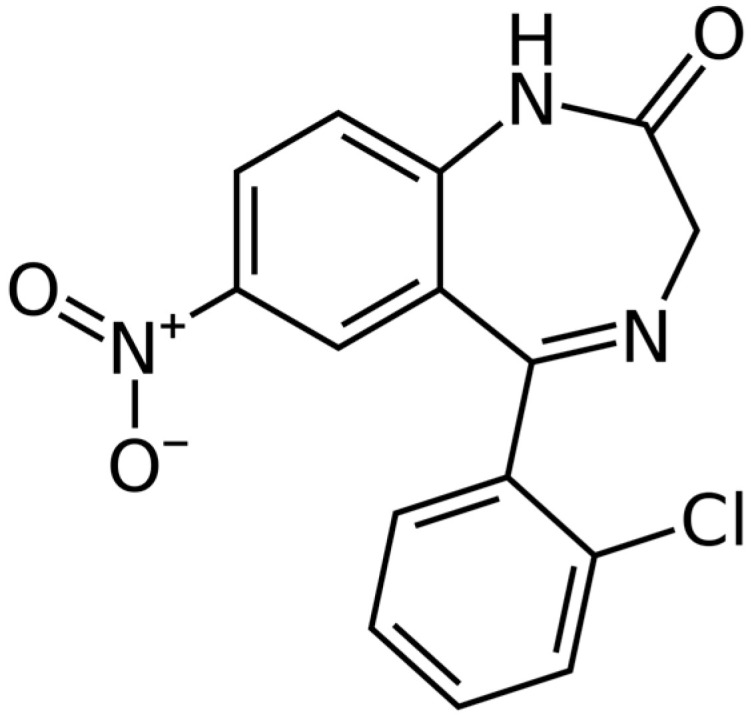
Structure of clonazepam.

**Figure 2 pharmaceutics-17-01302-f002:**
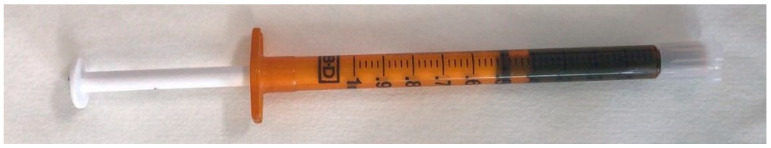
BD^TM^ Oral syringe loaded with 0.4 mL of clonazepam 2.5 mg/mL.

**Figure 3 pharmaceutics-17-01302-f003:**
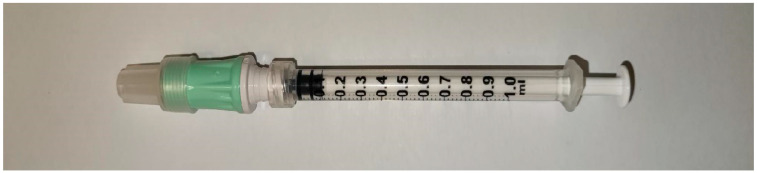
Nipro Luer lock syringe with closed safety system to load with 1 mL of clonazepam 1 mg/mL.

**Figure 4 pharmaceutics-17-01302-f004:**
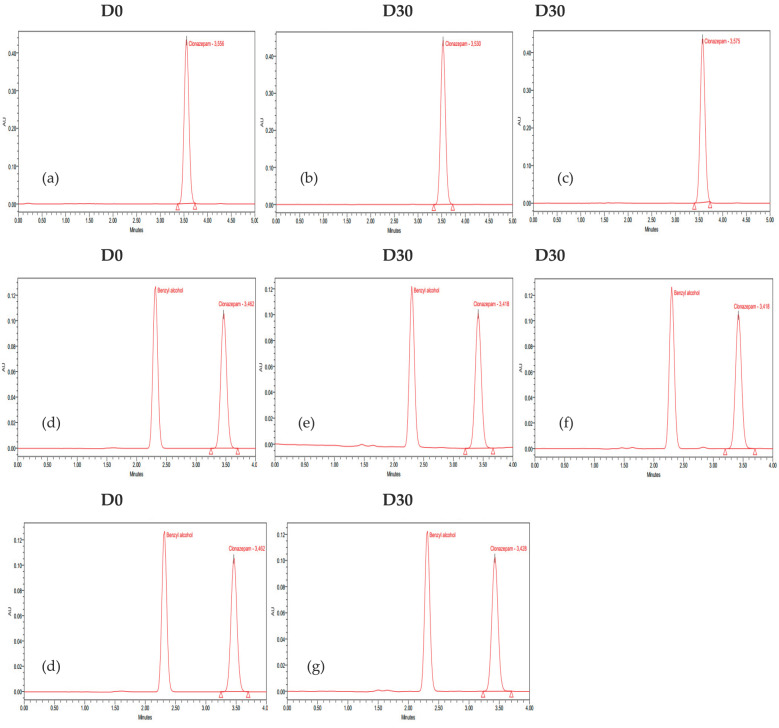
Chromatograms of clonazepam. Oral clonazepam 2.5 mg/mL solution in pre-filled syringes: (**a**) D0; (**b**) D30 room temperature; (**c**) refrigeration condition protected from light. Parenteral clonazepam 1 mg/mL solution in pre-filled syringes: (**d**) D0; (**e**) room temperature; (**f**) room temperature protected from light; (**g**) refrigeration condition protected from light.

**Figure 5 pharmaceutics-17-01302-f005:**
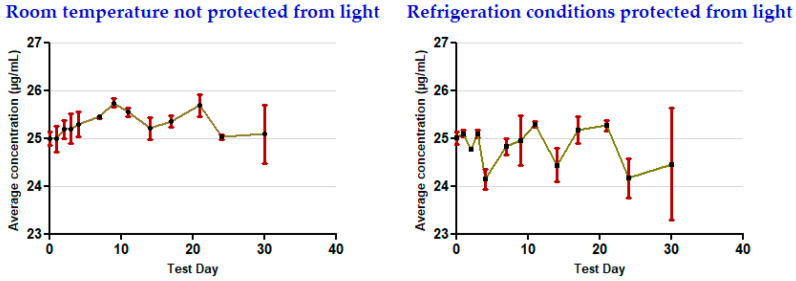
Evolution of clonazepam concentration during the study, related to oral clonazepam 2.5 mg/mL solution in pre-filled syringes (nominal concentration 25 µg/mL).

**Figure 6 pharmaceutics-17-01302-f006:**
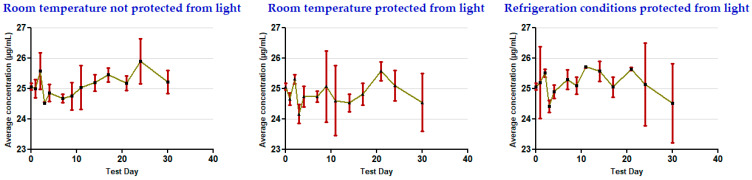
Evolution of clonazepam concentration during the study, related to parenteral clonazepam 1 mg/mL solution in pre-filled syringes (nominal concentration 25 µg/mL).

**Table 1 pharmaceutics-17-01302-t001:** Intra-day repeatability of clonazepam samples at 80, 100 and 120% of the target concentration of 25 µg/mL.

Theoretical Concentration (µg/mL)	20 (80%)	25 (100%)	30 (120%)
Mean (µg/mL)	20.04	24.99	30.04
SD	0.03	0.02	0.21
RSD%	0.15	0.09	0.68
Accuracy%	100.20	99.98	100.12

SD = Standard deviation; RSD%: Relative standard deviation.

**Table 2 pharmaceutics-17-01302-t002:** Inter-day repeatability of clonazepam samples at 80, 100 and 120% of the target concentration of 25 µg/mL.

Theoretical Concentration (µg/mL)	20 (80%)	25 (100%)	30 (120%)
Mean (µg/mL)	20.04	25.04	30.10
SD	0.07	0.06	0.21
RSD%	0.33	0.23	0.71
Accuracy%	100.21	100.15	100.32

SD = Standard deviation; RSD%: Relative standard deviation.

**Table 3 pharmaceutics-17-01302-t003:** Physicochemical results of oral clonazepam 2.5 mg/mL solution in pre-filled syringes.

	Room Temperature Not Protected from Light		Refrigeration Conditions Protected from Light	
	D0	D30	D0	D30
Average Recovery % of concentration	100	100.33 ± 0.01	100	97.82 ± 0.02
pH	4.63 ± 0.02	4.65 ± 0.06	4.65 ± 0.03	4.65 ± 0.05
Color	Blue	Blue	Blue	Blue
Crystals ≥10 µm/mL	0	0	0	0

Results expressed as mean ± SD (standard deviation) of triplicate determinations; D0 = Day 0 of the test; D30 = Day 30 of the test.

**Table 4 pharmaceutics-17-01302-t004:** Physicochemical results of parenteral clonazepam 1 mg/mL solution in pre-filled syringes.

		Room Temperature Not Protected from Light	Room Temperature Protected from Light	Refrigeration Conditions Protected from Light
	D0	D30	D30	D30
Average Recovery % of concentration	100	100.87 ± 0.01	98.14 ± 0.02	98.02 ± 0.02
pH	4.15 ± 0.08	4.27 ± 0.15	4.30 ± 0.07	4.17 ± 0.04
Color	Transparent	Transparent	Transparent	Transparent
Crystals ≥10 µm/mL	0	0	0	0

Results expressed as mean ± SD (standard deviation) of triplicate determinations; D0 = Day 0 of the test; D30 = Day 30 of the test.

## Data Availability

The data presented in this study are available on request from the corresponding author.

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
