# Peer review of "Stability Studies of Clonazepam 2.5 mg/mL Oral Solution and 1 mg/mL Parenteral Solution in Pre-Filled Polypropylene Syringes"

_pharmaceutics, 2025, doi:10.3390/pharmaceutics17101302_

Round 1
Reviewer 1 Report (Previous Reviewer 2)
Comments and Suggestions for Authors
The revised manuscript that discussed on the stability aspects of light-protected pre-filled PP syringes for the injectable formulations with fixed concentration of clonazepam oral solution may be acceptable.
Author Response
Comments1: The revised manuscript that discussed on the stability aspects of light-protected pre-filled PP syringes for the injectable formulations with fixed concentration of clonazepam oral solution may be acceptable.
Response1: Thank you very much for your help.
Reviewer 2 Report (New Reviewer)
Comments and Suggestions for Authors
The work presented by Ramizes et al. demonstrates the study of clonazepam stability after processing of the medicine in a hospital environment. The research is both interesting and important for highlighting the correct procedures in hospitals. However, the analytical HPLC method must be stability indicating to support the authors’ conclusions, and stress studies need to be carried out.
Despite its importance, the article is relatively simple for a high-standard and demanding journal such as Pharmaceutics, and it might be more suitable for submission to a journal of lower impact.
Author Response
Comments and Suggestions for Authors
The work presented by Ramirez et al. demonstrates the study of clonazepam stability after processing of the medicine in a hospital environment. The research is both interesting and important for highlighting the correct procedures in hospitals. However, the analytical HPLC method must be stability indicating to support the authors’ conclusions, and stress studies need to be carried out.
Despite its importance, the article is relatively simple for a high-standard and demanding journal such as Pharmaceutics, and it might be more suitable for submission to a journal of lower impact.
Response:
Dear Sir,
Normally, stress studies are performed during the development, registration, and validation of medicines, especially new ones. This is not our case. The medicine has been marketed for many years, but the advancement of knowledge has led us to use and handle medicines in a manner different from what we used to do. A clear example can be found with cytostatic drugs, which are now prepared in Pharmacy Departments under conditions unimaginable compared to those used thirty years ago, for example.
Pharmaceutical companies usually employ stress studies to support the design of accelerated and long-term studies, which is not our case, since our aim was not to determine the long-term stability of clonazepam concentrations studied in polypropylene syringes, but rather to obtain stability data that guarantee the possibility of making preparations at a certain frequency, for instance every 15 or 30 days, with full assurance, avoiding the need for daily preparations in Pharmacy Departments. The study we have carried out seeks to provide a practical solution to a real-world problem in the hospital setting. The analytical technique employed is considered appropriate and has been used in other studies referenced in our research. Moreover, it was validated according to ICH guidelines, demonstrating suitable accuracy, precision, and reproducibility.
We will review our manuscript and take into account your comments, making reference to the aspects you raise in your feedback. Accordingly, we have included the following statement in the Materials and Methods section, subsection 2.3 Chromatographic Method: "This chromatographic method had been previously validated in a published study, which assessed its stability by analyzing clonazepam samples subjected to heat, light, acidic, basic, and oxidative degradation. The study demonstrated that the presence of degradation products did not interfere with the detection of the intact drug.
Thank you very much for yours comments.
Sincerely
Reviewer 3 Report (New Reviewer)
Comments and Suggestions for Authors
This study evaluated the physicochemical stability of clonazepam solutions intended for oral (2.5 mg/mL) and parenteral (1 mg/mL) administration following repackaging in prefilled polypropylene (PP) syringes. The researchers found that both solutions remained stable for up to 30 days under the tested storage conditions, which included refrigeration and room temperature with and without light protection. Degradation of the active ingredient was less than 10%, and no changes in visual appearance, pH, or crystal formation were observed. These results suggest that centralizing the repackaging of clonazepam in PP syringes is a safe strategy for reducing drug handling by nursing staff and decreasing the risk of occupational exposure.
Overall, the paper is well-written. The authors provide context by explaining the problems posed by clonazepam. It is a dangerous drug that requires special precautions for personnel at reproductive risk. A key aspect of the manuscript is its identification of a significant knowledge gap, which focuses the study on the absence of stability studies for clonazepam in pre-filled PP syringes. This establishes the work's relevance.
The research provides significant data supporting the proposals of the hospital's Pharmacy and Therapeutic Commission and Occupational Risk Prevention Service. While the data is compelling, the discussion is clear, and the conclusions are relevant, the following recommendations are suggested.
- Present graphs showing the behavior of drug concentration over time in each of the study conditions, along with their respective error bars.
- Analyzing the degradation kinetics would be relevant.
- An important factor is analyzing degradation products. Although the research does not address them, reviewing the literature would be important.
Author Response
This study evaluated the physicochemical stability of clonazepam solutions intended for oral (2.5 mg/mL) and parenteral (1 mg/mL) administration following
repackaging in prefilled polypropylene (PP) syringes. The researchers found that both solutions remained stable for up to 30 days under the tested storage
conditions, which included refrigeration and room temperature with and without light protection. Degradation of the active ingredient was less than 10%, and no
changes in visual appearance, pH, or crystal formation were observed. These results suggest that centralizing the repackaging of clonazepam in PP syringes is a
safe strategy for reducing drug handling by nursing staff and decreasing the risk of occupational exposure.
Overall, the paper is well-written. The authors provide context by explaining the problems posed by clonazepam. It is a dangerous drug that requires special
precautions for personnel at reproductive risk. A key aspect of the manuscript is its identification of a significant knowledge gap, which focuses the study
on the absence of stability studies for clonazepam in pre-filled PP syringes. This establishes the work's relevance.
The research provides significant data supporting the proposals of the hospital's Pharmacy and Therapeutic Commission and Occupational Risk Prevention Service.
While the data is compelling, the discussion is clear, and the conclusions are relevant, the following recommendations are suggested.
Present graphs showing the behavior of drug concentration over time in each of the study conditions, along with their respective error bars.
Analyzing the degradation kinetics would be relevant.
An important factor is analyzing degradation products. Although the research does not address them, reviewing the literature would be important.
Dear Sir,
We have introduced in the manuscript, results section, two news graphics with drug concentration over time in each of de study conditions (Figure 5 and Figure 6).
We have included the following statement in the Materials and Methods section, subsection 2.3 Chromatographic Method:
This chromatographic method had been previously validated in a published study, which assessed its stability by analyzing clonazepam samples subjected to heat,
light, acidic, basic, and oxidative degradation. The study demonstrated that the presence of degradation products did not interfere with the
detection of the intact drug.
We have included the following paragraphs in the discussions section:
The HPLC method used for the determination of clonazepam was linear, precise, ac-curate, repeatable, and reproducible, and was validated in accordance with ICH guide-lines. This method was demonstrated to be robust and capable of reliably resolving clonazepam in the presence of its degradation products [17], consistent with the findings of other published studies [20,24].
No studies have evaluated the degradation kinetics of clonazepam in an oral solution at a concentration of 2.5 mg/mL or in a parenteral solution at 1 mg/mL, both packaged in polypropylene syringes. One study assessed the stability of an acidic clonazepam solution at 2 mg/mL in glass ampoules, concluding that clonazepam degradation follows first-order kinetics [25]; similarly, another study investigated the stability of clonazepam tablets in combination with escitalopram [24]. As already mentioned in the introduction, both clonazepam concentrations analyzed in this study have a manufacturer-reported shelf life of three years when stored in primary glass containers under the recommended storage conditions, suggesting that both are highly stable. The aim of this study was not to determine whether repackaging commercial clonazepam solutions in polypropylene sy-ringes could affect or alter the degradation kinetics of the active ingredient, although con-ducting such an investigation would be advisable to further strengthen the conclusions.
Reviewer 4 Report (New Reviewer)
Comments and Suggestions for Authors
11.09.2025
A review to evaluate its suitability for publication Type of manuscript:
Article
Title: Stability studies of Clonazepam 2.5 mg/mL oral solution and 1 mg/mL parenteral solution in pre-filled polypropylene syringes
Authors: Juan C. Ruiz Ramirez*, Icram Talsi Hamdani, Laura Bermúdez Gazquez, Alice Charlotte Viney, José M. Alonso Herreros
The authors' manuscript is devoted to studying the stability of aqueous solutions of Clonazepam, packed in polypropylene syringes, in order to avoid direct contact with the substance causing skin and mucous reactions of the body. In fact, we are talking about studying the aging processes in natural conditions, which is very important for the tasks of pharmacy. In this regard, this manuscript is devoted to a practical task implemented in a medical institution.
The manuscript is written in the competent language of an analytical chemist, with a clear description of the relevance, the goal of the work, a description of the objects and methods, and a brief but structured presentation of the results obtained.
The following questions arose while reading the manuscript:
- It would be desirable to provide the structural formula of clonazepam, a typical antidepressant, indicating those of its features that determine its ability to be destroyed, as well as its unsafety (cyclic amide, covalently bonded chlorine atom, aromatic nitro group..)
- Were the authors guided by the existing «NOTE FOR GUIDANCE ON EVALUATION OF STABILITY DATA» European Medicines Agency protocol https://www.ema.europa.eu/en/ich-q1e-evaluation-stability-data-scientific-guideline? Since the assessment of stability and aging is a trivial event in pharmacy, for which there is already a ready-made solution.
I hope that the answers to these questions will help the authors in their current work and in future research.
Respectfully, reviewer
Author Response
Dear Sir,
We have introduced in the manuscript, introduction section, one graphics with the structure of clonazepam (Figure1).
In addition, we have added one paragrah in the introduction section:
The aromatic nitro group, the cyclic amide, and the covalently bonded chlorine atom within the clonazepam molecule can influence its chemical stability,
both by environmental factors and storage conditions, while also impacting its pharmacological properties and toxicological profile. The nitro group may be
susceptible to reduction reactions, potentially affecting the drug’s toxicity and safety. Under extreme conditions, it could contribute to the formation of
toxic degradation products. The cyclic amide imparts molecular rigidity, enhancing resistance to chemical degradation and hydrolysis; however, under extreme
temperature and pH conditions, this cyclic group may break, facilitating degradation of the active pharmaceutical ingredient. The covalently bonded chlorine
atom is chemically stable, conferring additional chemical resistance, although it can serve as a reactive site in photochemical or oxidative degradation
processes; moreover, it contributes to increasing the molecule’s lipophilicity, thereby affecting its metabolism and distribution.
Thank you very much for yours comments.
Sincerely
Reviewer 5 Report (New Reviewer)
Comments and Suggestions for Authors
- In the introduction, line 80, there appears to be a typo (“?” should be replaced with “.”). Please correct this.
- Did you evaluate the homogeneity of the formulation after 30 days of storage in the syringe (ex. checking for separation or layering)?
- Was any assessment performed to determine whether the drug interacted with the syringe plunger, or whether leakage or blockage occurred after 30 days of storage?
- For crystallization, you mention only visual observation. Very small crystals may not be visible to the naked eye. Did you perform microscopic analysis or turbidity/light scattering measurements to quantify the possible formation of microcrystals?
- It would strengthen the study to include PXRD or DSC analysis to determine whether the crystalline state of the drug was maintained or altered during storage.
- While you performed an assay for drug concentration to evaluate stability, can you clarify why no dissolution/release testing was performed?
- You mentioned that clonazepam for oral use is normally stored in glass containers. Why was a glass syringe not used as a control to allow comparison with the PP syringe?
Author Response
Comments and Suggestions for Authors
In the introduction, line 80, there appears to be a typo (“?” should be replaced with “.”). Please correct this.
Did you evaluate the homogeneity of the formulation after 30 days of storage in the syringe (ex. checking for separation or layering)?
Was any assessment performed to determine whether the drug interacted with the syringe plunger, or whether leakage or blockage occurred after 30 days of storage?
For crystallization, you mention only visual observation. Very small crystals may not be visible to the naked eye. Did you perform microscopic analysis or turbidity/light scattering measurements to quantify the possible formation of microcrystals?
It would strengthen the study to include PXRD or DSC analysis to determine whether the crystalline state of the drug was maintained or altered during storage.
While you performed an assay for drug concentration to evaluate stability, can you clarify why no dissolution/release testing was performed?
You mentioned that clonazepam for oral use is normally stored in glass containers. Why was a glass syringe not used as a control to allow comparison with the PP syringe?
Dear Sir,
We have corrected the typo "?" and replaced with "."
Did you evaluate the homogeneity of the formulation after 30 days of storage in the syringe (ex. checking for separation or layering)?
We have not evaluated the homogeneity cheking for separation o layering, but between the methods and procedures that can be applied to evaluate the homogeneity
of a pharmaceutical solution, the HPLC quantitative analytical assay is a standard.
Was any assessment performed to determine whether the drug interacted with the syringe plunger, or whether leakage or blockage occurred after 30 days of storage?
Each test day we check the posibility of leakage or blockage. We are modify the subsection 2.4 Physical stability incorporating this feature. We have written about it
in the discussion section.
For crystallization, you mention only visual observation. Very small crystals may not be visible to the naked eye. Did you perform microscopic analysis or turbidity/light
scattering measurements to quantify the possible formation of microcrystals?
In the subsection 2.4. Physical stability we comment it:"A visual inspection booth with both a black/white background [12] and bright-field, and a SediMAX2TM phase contrast microscope (77 Elektronika, Budapest, -Hungary-),
were used to determine the presence of particles and crystals. Therefore, we have used phase contrast microscopy.
It would strengthen the study to include PXRD or DSC analysis to determine whether the crystalline state of the drug was maintained or altered during storage.
You are right, but these techniques are not available in our laboratory. We will writte a comment in the discussion section.
While you performed an assay for drug concentration to evaluate stability, can you clarify why no dissolution/release testing was performed?
Dissolution testing as per pharmacopeial requirements is primarily mandatory for solid oral dosage forms such as tablets and capsules, including immediate-release,
modified-release, delayed-release, and enteric-coated formulations. These tests are generally not required for liquid dosage forms, such as true solutions or syrups,
since the active pharmaceutical ingredient is already dissolved. The obligation to conduct dissolution testing applies specifically to dosage forms where it is
essential to ensure the adequate release and dissolution of the active ingredient in the body, thereby guaranteeing its bioavailability and therapeutic efficacy.
For that reason we did not do dissolution testing.
You mentioned that clonazepam for oral use is normally stored in glass containers. Why was a glass syringe not used as a control to allow comparison with the PP
syringe?
Currently, the use of glass syringes in hospital settings is uncommon. Polypropylene or polycarbonate syringes are preferred due to their favorable compatibility
with pharmaceuticals, as mentioned in the introduction of this work, and because they are more cost-effective than glass. For this reason, glass syringes were not
utilized in our study. However, the use of a control with glass syringes, as you suggested, is considered a valid approach, and we will take this recommendation
into account for future investigations.
Thank you very much for yours comments.
Sincerely
Round 2
Reviewer 2 Report (New Reviewer)
Comments and Suggestions for Authors
The authors have answered the main issue I've raised, the stability indicating capability of the method. Although I still think its a relatively simple research for a journal as Pharmaceutics, now it could be accepted for publication.
Reviewer 3 Report (New Reviewer)
Comments and Suggestions for Authors
The authors made the suggested changes
Reviewer 5 Report (New Reviewer)
Comments and Suggestions for Authors
The authors have adequately addressed all of my previous comments.
This manuscript is a resubmission of an earlier submission. The following is a list of the peer review reports and author responses from that submission.
Round 1
Reviewer 1 Report
Comments and Suggestions for Authors
This study provides clinically valuable stability data for clonazepam repackaged in pre-filled syringes (PFS), addressing occupational safety concerns. However, there have been some research works on stability evaluation of clonazepam. Two questions should be considered:
1. What is the chemical stability of clonazepam itself under different conditions?
2. Do pre-filled syringes (PFS) affect clonazepam's stability? Why?
If clonazepam is inherently stable and PFS do not degrade it (as concluded), then the research primarily confirms expected behavior in a new format. While this has practical value for occupational safety (enabling PFS use), it offers limited novel scientific insight. Therefore, I am negative to recommend this work to Pharmaceutics.
Author Response
Comments 1: What is the chemical stability of clonazepam itself under different conditions?
In Spain, there are two presentations of clonazepam commercialized in solution: oral drops (2.5 mg/mL) and parenteral (1 mg/mL). The oral solution is packaged in a topaz glass bottle and the parenteral in a topaz glass ampoule as primary packaging. The conservation conditions of two presentations are temperature equal o less than 30ºC in its secondary packaging to protect from the light. In these conditions, both have a three-year shelf life if unopened.
Only, one study of clonazepam solution 0.5 mg/mL in water for injection, packaged in polypropylene (PP) syringes has been published, with the result of 48 hours of stability, at 25º C, not protected from light.
We have incorpórated these questions into introduction section of the manuscript.
Comments 2: Do pre-filled syringes (PFS) affect clonazepam's stability? Why?
PP is considered one of the most inert plastics and with the lowest tendency to adsorb drugs; in practice, adsorption to PP is usually low, but it can be significant in drugs of low concentration, of protein or peptide nature, or of high unit value, such as certain radiopharmaceuticals or insulin. In addition, many studies have shown that the physi-cochemical stability of most drugs is adequate in PP syringes, with no appreciable losses over periods ranging from several days to weeks, depending on the drug and the storage conditions (refrigeration, darkness). PP offers low permeability to water vapor and is relatively impermeable to oxygen, which helps protect the solution from degradation by oxidation or hydrolysis. However, its gas barrier is not as hermetic as that of glass, so in long-term storage situations the shelf life of very sensitive solutions could be affected. Pharmaceutical-grade PP syringes minimize the presence of particles and the migration of additives.
Therefore the stability of clonazepam solutions (2,5 mg/mL and 1 mg/mL) could be affected by PP under different storage conditions.
We have incorpórated these questions into the introduction section of the manuscript.
Reviewer 2 Report
Comments and Suggestions for Authors
The manuscript discussed on the 1 month stability of injectable formulation of Clonazepam in prefilled syringes. I have objections on the novelty of the work than the reported ones discussed in the section 4. If the concentration is more like claimed in this case, possibility of stability will be also more.
"studies of Clonazepam 1 mg/mL (parenteral use) and 2.5 mg/mL (oral use) in solution" what is the solution?
Author Response
Comments 1: The manuscript discussed on the 1 month stability of injectable formulation of Clonazepam in prefilled syringes. I have objections on the novelty of the work than the reported ones discussed in the section 4. If the concentration is more like claimed in this case, possibility of stability will be also more.
The relationship between the concentration of a drug in solution and its stability is not generally directly proportional, but rather depends on multiple chemical, physical, and environmental factors surrounding the drug.The chemical stability of a drug in solution can be affected by concentration, but the relationship is not always linear or direct. In certain cases, higher concentrations can favor stability by reducing the proportion of water, but in other cases, they can increase degradation due to chemical or physical reactions that depend on the active ingredient concentration and on other factors such as pH, temperature, and the presence of ions or catalysts. Stability also depends on degradation kinetics. This determines how the drug concentration changes over time and how it degrades, regardless of whether its concentration is low or high. For example, in some cases the degradation rate is proportional to the concentration, and in others it is not. For these reasons the stability must be specifically studied for each formulation, as behavior can vary depending on the drug and its environment.
We have incorpórated these questions into the introduction section of the manuscript.
Comments 2 "studies of Clonazepam 1 mg/mL (parenteral use) and 2.5 mg/mL (oral use) in solution" what is the solution?
In this context, "solution" refers to a homogeneous liquid preparation in which the active ingredient (Clonazepam) is dissolved in a suitable liquid (solvent) for its administration, either by the parenteral (injection) or oral route.
Reviewer 3 Report
Comments and Suggestions for Authors
The current article focusing on evaluating the stability of clonazepam solution (oral and injectable) needs to be further revised for additional information. Authors are suggested to address the below comments:
- In the abstract instead of just mentioning as hazardous drug, please add a note in what aspects it was considered hazardous such as safety?
- Line 30: update as protected and unprotected if both the cases were studied
- Line 32: Instead of loss, update it as degradation or update it as “the recovery was greater than 90%”
- The abstract needs to be revised for more professional and scientific presentation of the content.
- Introduction needs to be improved and provide a detailed note why the drug was considered hazardous and also provide a detailed physicochemical property of the drug along with potential stability challenges
- In the tables please make sure the “.” Were presented as “,”
- Can authors please explain why the samples “protected from light” have resulted 2% drop in assay, though its not significant drop.
- How can authors assure the sterility of injectable solution?
- Are these solutions directly received from the manufacturer or were they reconstituted before filing into the syringes?
- Please make a note of the formulation compositions in the introduction section of the article.
Needs improvement
Author Response
The current article focusing on evaluating the stability of clonazepam solution (oral and injectable) needs to be further revised for additional information. Authors are suggested to address the below comments:
Comment 1: In the abstract instead of just mentioning as hazardous drug, please add a note in what aspects it was considered hazardous such as safety?
We have incorpórated these questions into the manuscript in the introduction section.
Comment 2: Line 30: update as protected and unprotected if both the cases were studied
We have update this line as: Two groups of syringes were stored at either controlled room temperature (25ºC) protected and unprotected from light....
Comment 3: Line 32: Instead of loss, update it as degradation or update it as “the recovery was greater than 90%”
We have changed the word loss by degradation
Comment 4: The abstract needs to be revised for more professional and scientific presentation of the content.
We have revised the abstract in this way.
Comment 5: Introduction needs to be improved and provide a detailed note why the drug was considered hazardous and also provide a detailed physicochemical property of the drug along with potential stability challenges
Bases on your recommendations, we have revised the introduction in this way.
Comment 6: In the tables please make sure the “.” Were presented as “,”
We have changed "," by "." in the tables.
Comment 7: Can authors please explain why the samples “protected from light” have resulted 2% drop in assay, though its not significant drop.
In a drug stability study, a 2% decrease from the initial concentration during the study period is generally considered within an acceptable margin and does not necessarily indicate a significant instability issue. Stability and regulatory guidelines usually allow small variations in the active ingredient concentration as normal, understanding that minimal losses may be due to physical or experimental factors. Stability and regulatory guidelines usually allow small variations in the active ingredient concentration as normal, understanding that minimal losses may be due to physical or experimental factors. If the loss is less than 5%, in most cases it is not reviewed as a stability failure unless accompanied by other physical, chemical, or microbiological changes. The evaluation is complemented by degradation analysis (degradation products), changes in pH, appearance, and other parameters to ensure that the drug remains stable and safe for use.
Comment 8: How can authors assure the sterility of injectable solution?
This study does not evaluate the microbiological stability of clonazepam 1 mg/mL parenteral solución. However, the fact that it was repacked in syringes under aseptic conditions (vertical laminar Flow cabinet in a controlled enviroment) and that the comercial solution used contains benzy alcohol (30 mg/mL) as a preservative are crucial in ensuring microbiological stability.
We will discuss this issue in the discussion section.
Comment 9: Are these solutions directly received from the manufacturer or were they reconstituted before filing into the syringes?does
These solutions are directly received from the manufacturer, because are commercial solutions.
Comment 10: Please make a note of the formulation compositions in the introduction section of the article.
We will make a note of the formution compositions in the introduction section.